# Probing the crystallographic orientation of two-dimensional atomic crystals with supramolecular self-assembly

Jinghui Wang[1], Hongde Yu[1], Xu Zhou[2], Xiaozhi Liu[3], Renjie Zhang[4], Zhixing Lu[1], Jingying Zheng[1], Lin Gu[3], Kaihui Liu[2], Dong Wang[1] & Liying Jiao[1]

Probing the crystallographic orientation of two-dimensional (2D) materials is essential to understand and engineer their properties. However, the nondestructive identification of the lattice orientations of various 2D materials remains a challenge due to their very thin nature. Here, we identify the crystallographic structures of various 2D atomic crystals using molecules as probes by utilizing orientation-dependent molecule–substrate interactions. We discover that the periodic atomic packing of 2D materials guides oleamide molecules to assemble into quasi-one-dimensional nanoribbons with specific alignments which precisely indicate the lattice orientations of the underlying materials. Using oleamide molecules as probes, we successfully identify the crystallographic orientations of ~12 different 2D materials without degrading their intrinsic properties. Our findings allow for the nondestructive identification of the lattice structure of various 2D atomic crystals and shed light on the functionalization of these 2D materials with supramolecular assembly.

[1] Key Laboratory of Organic Optoelectronics and Molecular Engineering of the Ministry of Education, Department of Chemistry, Tsinghua University, Beijing 100084, China. [2] State Key Laboratory for Mesoscopic Physics, Collaborative Innovation Center of Quantum Matter, School of Physics, Academy for Advanced Interdisciplinary Studies, Center for Nanochemisty, Peking University, Beijing 100871, China. [3] Beijing National Laboratory for Condensed Matter Physics, Institute of Physics, Chinese Academy of Sciences, Beijing 100190, China. [4] Key Laboratory of Colloid and Interface Chemistry of the Ministry of Education, Shandong University, Jinan 250100, China. Jinghui Wang and Hongde Yu contributed equally to this work. Correspondence and requests for materials should be addressed to D.W. (email: dong913@mail.tsinghua.edu.cn) or to L.J. (email: lyjiao@mail.tsinghua.edu.cn)

Two-dimensional (2D) atomic crystals, endowed with unique ultrathin planar and well-ordered atomic structures, have shown a broad range of attractive properties, which are dramatically different from their bulk counterparts[1, 2]. Among these properties, electrical transport[3, 4], thermal conductance[5, 6], piezoelectric characteristics[7], optical properties[3] and so on are significantly dependent on their 2D crystallographic orientations, especially for those with in-plane anisotropic structures such as black phosphorus[3] and ReS$_2$[4]. Besides these intrinsic properties, modulating the properties of 2D atomic crystals by strain, stacking and size-confinement is also highly dependent on their pristine lattice orientations as indicated by various theoretical and experimental studies[8–10]. Therefore, probing the crystallographic orientation of these materials is essential for understanding and controllably tuning their properties in an expanding array of 2D atomic materials. The most straightforward characterization is to directly image the atomic structure of 2D materials with scanning transmission electron microscope (STEM)[11] or scanning tunneling microscope (STM)[12]. However, the acquisition of atomically resolved images by both STEM and STM has critical requirements on sample preparation, equipment and imaging skills, which makes these approaches very expensive and low throughput. More importantly, the imaged samples are located on specialized substrates such as transmission electron microscope grids or conducting substrates, which hinders their further applications. Recently, orientation-dependent intensities of second harmonic generation (SHG) and polarized Raman spectroscopy have been observed in several 2D atomic crystals, providing an alternative approach for identifying lattice orientation of several 2D materials, such as odd-layered BN[13], MoS$_2$[14], WS$_2$[15] and GaSe[16] by SHG and black phosphorus[17] and ReS$_2$ (ReSe$_2$)[18–20] by Raman spectroscopy. Unfortunately, these spectroscopic approaches have strict requirements on the symmetry and structures of the materials and thus cannot be adapted to identify the crystallographic orientations of a large variety of 2D materials. A universal and nondestructive approach for rapidly probing the lattice orientation of various 2D atomic crystals is therefore crucial to advance the field of 2D atomic crystals.

Organizing molecules into geometric order on surfaces has attracted extensive interest for constructing nano-devices and nanostructured materials using bottom-up approaches[21]. It is well-accepted that molecule–substrate interactions significantly affect the adsorption of single molecules on solid surfaces and thus can guide the orientation of molecules[22]. Much effort has been made to investigate the effects of substrate surface structure on the assembly of molecules[23, 24]. However, whether the geometry of the resulting supramolecular structures can reveal the atomic structures of the substrate has not been explored.

We expect that periodic atomic arrangements of 2D atomic crystals result in periodic molecule–substrate interactions in the 2D plane and thus provide a unique landscape for aligning molecules. Proceeding from this point, we explore the self-assembly of molecules on 2D atomic crystals from both experimental and theoretical aspects and successfully develop a universal, simple and nondestructive approach to identify the lattice structures of various 2D atomic crystals using molecules as probes.

## Results

**Orientation-selected assembly of oleamide molecules on 2D atomic crystals**. The well-ordered assembly of various molecules has been explored mainly on the surfaces of metals and graphite by STM imaging[22]. Among these molecules, here, we choose

oleamide as the probe to identify the lattice orientations of 2D atomic crystals as these molecules can form anisotropic quasi-one-dimensional (1D) structures driven by intermolecular hydrogen bonds on highly oriented pyrolytic graphite[25, 26], which is highly desirable for indicating the lattice orientation of the substrates. The assembly process of oleamide onto 2D atomic crystals simply involves two steps: spin-coat a drop of dilute oleamide solution in chloroform onto 2D materials and then anneal the samples at 60 °C for 30 min (see Methods for experimental details). After the assembly, dense nanoribbons (500 ± 200 nm, 50 ± 20 nm and 5 ± 4 nm in length, width and height, respectively, as shown in Supplementary Fig. 1) were observed on various hexagonal 2D atomic crystals such as mechanically exfoliated graphene, h-BN, MoS$_2$, MoSe$_2$, MoTe$_2$, WS$_2$, WSe$_2$, NbSe$_2$, TaS$_2$ and so on (Fig. 1b–g and Supplementary Fig. 2), and these nanoribbons aligned along three directions with rotations of ~120° (Supplementary Fig. 3) as schematically illustrated in Fig. 1a over a large area (Supplementary Figs. 4 and 5). In addition to these hexagonal crystals, nanoribbons on black phosphorus with tetragonal symmetry and ReS$_2$ with distorted 1T structure also displayed specific orientations along two and one directions, respectively (Fig. 1h, i). We ruled out the possible formation of oleamide nanoribbons in solution through dynamic light scattering (DLS) measurements (Supplementary Fig. 6) and a control experiment on the self-assembly of 5, 10, 15, 20-tetraphenyl-21H, 23H-porphine cobalt (II) (CoTPP) on MoS$_2$ (Supplementary Figs. 6 and 7), and therefore, confirmed that oleamide nanoribbons were formed on the surface of 2D materials instead of in solution. The dramatically different orientations of oleamide nanoribbons on 2D atomic crystals with varied crystallographic structures suggest that the alignment of nanoribbons is very likely to be guided by the atomic structures of the underlying substrates.

**Orientation-correlations between oleamide nanoribbons and 2D materials**. To correlate the orientations of the oleamide nanoribbons with the lattice structure of 2D atomic crystals, we characterized several 2D atomic crystals with oleamide assembly by combining STEM, SHG and polarized Raman spectroscopy. We first recorded the geometry of MoS$_2$ flakes with oleamide assembly by atomic force microscope (AFM) and then selectively transferred the same MoS$_2$ flake to a holey carbon grid for atomically resolved STEM imaging (Fig. 2a, b, see Supplementary Methods for experimental details). The correlation between the orientations of the nanoribbons and the MoS$_2$ atomic structures can be accurately affirmed by overlapping the geometric images taken by AFM and STEM (Supplementary Fig. 8a, b) and the results confirmed that oleamide nanoribbons oriented along the zigzag directions of MoS$_2$ lattice (see Supplementary Note 1 for more details). Besides STEM imaging, SHG was also utilized to characterize the lattice orientations of MoS$_2$ with oleamide nanoribbons. The angle-resolved SHG intensity measured on monolayer MoS$_2$ (Fig. 2c) with oleamide nanoribbons shows a clear six-petal pattern and the maximum petal direction points to the armchair lattice orientations of MoS$_2$ (Fig. 2d)[14]. Hence, the zigzag lattice orientations of the MoS$_2$ monolayer are determined by rotating the armchair directions by 30°, which are exactly the same as the orientations of the nanoribbons shown in Fig. 2c, again, confirming that oleamide nanoribbons align along the zigzag orientations of MoS$_2$. Besides MoS$_2$, we conducted SHG measurement on WSe$_2$ with oleamide nanoribbons and obtained the same conclusion (Supplementary Fig. 9).

Besides hexagonal 2D materials with threefold symmetry, we also explored the correlations between the nanoribbon orientations and the lattice structures of 2D black phosphorus

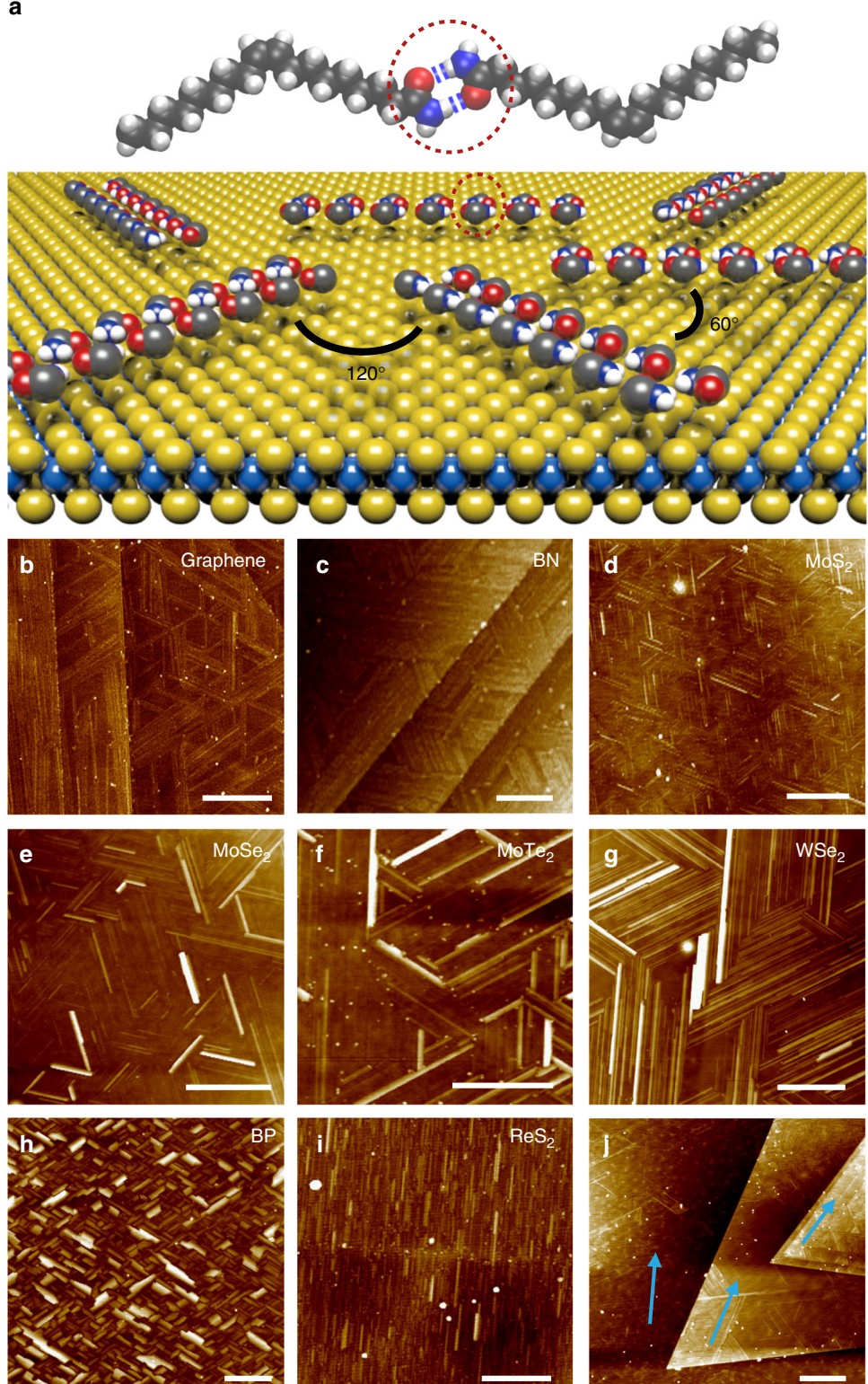

**Fig. 1** Self-assembly of oleamide on various 2D atomic crystals. **a** Schematic representation of oleamide self-assembly on 2D atomic crystals along specific orientations. For clarity, only the amide groups are displayed and the alkyl chains of oleamide are not shown except for the topmost oleamide dimer which shows the complete molecular structures. Hydrogen bonds are indicated with *blue dashed* lines. *White*, *gray*, *blue* and *red* balls represent H, C, N and O atoms, respectively. **b–i** AFM images of oleamide nanoribbons on graphene, *h*-BN, $MoS_2$, $MoSe_2$, $MoTe_2$, $WSe_2$, *black* phosphorus (BP) and $ReS_2$, respectively. **j** Nanoribbons on CVD-grown multilayer $MoS_2$. *Blue arrows* represent the orientations of nanoribbons on varied layers. All *scale bars* are 500 nm, except for that in **c** which is 200 nm

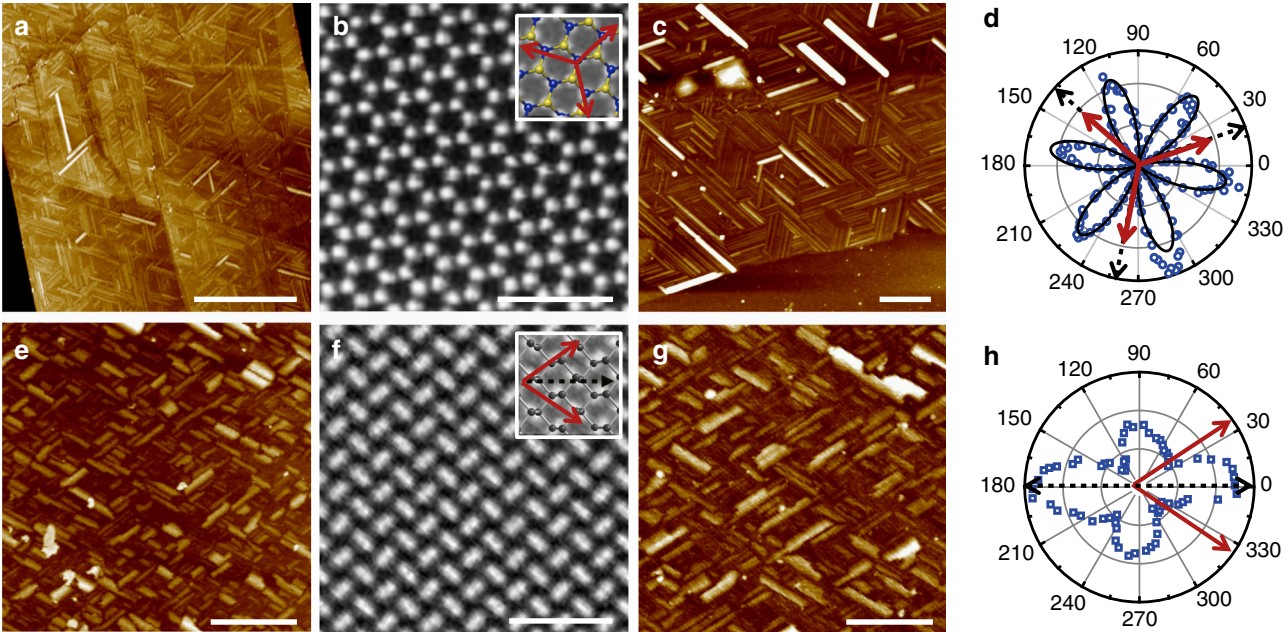

**Fig. 2** Correlations between the orientations of oleamide nanoribbons and the lattice of underlying 2D materials. **a**, **b** AFM image and Fourier filtered STEM image of the same few-layer $MoS_2$ flake with oleamide nanoribbons. *Inset* Schematic representation of top view of $MoS_2$ atomic structures. *Red arrows* correspond to the orientations of nanoribbons on this flake. **c**, **d** AFM image and polarization angle-dependent SHG intensity plot of the same monolayer $MoS_2$ flake with oleamide nanoribbons. The *black dashed* arrows indicate the zigzag directions of this $MoS_2$ flakes determined by SHG and the *red arrows* correspond to the orientations of nanoribbons on this flake. **e**, **f** AFM image and Fourier filtered STEM image of the same multilayer black phosphorus flake with oleamide nanoribbons. *Inset* Schematic representation of top view of the atomic structure of black phosphorus. *Red arrows* correspond to the orientations of nanoribbons. The *black dashed arrow* indicates the armchair direction of this flake. **g**, **h** AFM image and angle-dependent polarized Raman intensity plot of the same multilayer black phosphorus flake with oleamide nanoribbons. The *black dashed arrow* indicates the armchair direction of this flake determined by Raman measurements and the *red arrows* indicate the orientations of the nanoribbons on this flake. *Scale bars* for **a**, **c**, **e** and **g** are 500 nm and for **b**, **f** are 1 nm

with tetragonal symmetry and $ReS_2$ with triclinic symmetry (Fig. 2e and Supplementary Fig. 10c) with atomically resolved STEM and angle-resolved polarized Raman spectroscopy. Following the similar STEM imaging process of $MoS_2$, the armchair orientation of black phosphorus is determined to be along the bisector of the nanoribbon directions, with rotations of 33° to each nanoribbon orientations (Fig. 2e, f). Polarized Raman spectroscopy measurements on 2D black phosphorus with oleamide nanoribbons also revealed the same relationship between the orientations of nanoribbons and the armchair direction of black phosphorus (Fig. 2g, h)[27]. Likewise, the nanoribbons assembled on $ReS_2$ were verified to be aligned along the direction of Re chains (zigzag lattice structure) using angle-resolved polarized Raman spectroscopy (Supplementary Fig. 10)[18]. So far, we have demonstrated that the preferred orientations of oleamide nanoribbons can be well-correlated with the lattice structure of 2D atomic crystals experimentally.

**Theoretical calculations on the surface-templated assembly of oleamide.** Next, we investigated the origins for the preferred orientations of oleamide nanoribbons on 2D atomic crystals from theoretical aspects. The orientation of an oleamide molecule on 2D atomic crystals is dictated by the van der Waals interactions between adsorbate and substrate. So we first calculated the rotation potential energy curve of a single oleamide molecule on monolayer $MoS_2$ with only Grimme's D3 dispersion parameters (Supplementary Fig. 11) and the adsorption energy of a single oleamide molecule aligned along either zigzag or armchair direction of monolayer $MoS_2$ by the DFT-D3 method[28] (see Supplementary Methods for calculation details). For each alignment, we performed 2D potential energy surface (PES) scan

to ascertain the most favorable adsorption position by accounting for only dispersion interactions between oleamide and $MoS_2$. The 2D PES for zigzag aligned oleamide and its 1D projection along the $x$ (zigzag) and $y$ (armchair) directions respectively have been shown in Fig. 3a, b. The energy valleys in Fig. 3a exhibit the same hexagonal symmetry as the underlying lattice of $MoS_2$. Our DFT-D3 calculations of the adsorption energy show that the zigzag aligned oleamide is favored by 0.11 eV as compared to the armchair aligned one. Because the interactions between oleamide molecules and the surface are stronger than the intermolecular interactions between oleamide molecules[26], the surface lattice will template the formation of nanostructures that cannot be formed in solution[25] (Supplementary Fig. 6). Thus the adsorbate-substrate interactions dictate the assembly of oleamide molecules on the surface of 2D atomic crystals and the tendency to achieve maximum H-bonding between –C=O and -$NH_2$ groups of adjacent molecules servers as secondary interactions[26]. Based on the thermodynamically most stable adsorption position of individual zigzag aligned oleamide, we constructed the assembly pattern of oleamide molecules on $MoS_2$ (Fig. 3c). In this pattern, oleamide molecules not only form H-bonded dimers in a head-to-head configuration but also H-bonded nanoribbons in a side-by-side configuration. The orientations of nanoribbons are along the zigzag lattice directions of the underlying $MoS_2$, as also clearly seen from Fig. 3a. Our calculations also confirm that oleamide molecules form H-bonded nanoribbons on other hexagonal 2D atomic crystals including graphene, BN, $MoSe_2$, $MoTe_2$ and $WS_2$, along the zigzag lattice directions (Supplementary Fig. 12 and Supplementary Note 2). The first molecular layer of nanoribbons will in turn template the second layer assembly, and so on. The rotation and 2D PES scan showed that the second layer oleamide

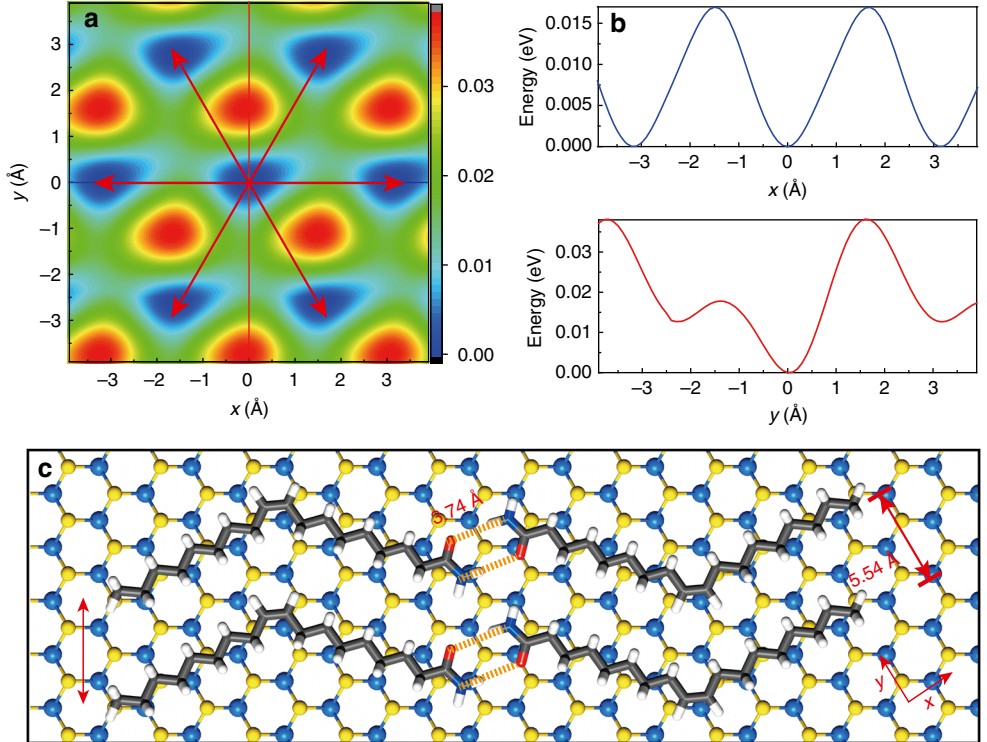

**Fig. 3** Theoretical calculations of oleamide assembly on monolayer MoS$_2$. **a** 2D potential energy surface (PES) of a single oleamide molecule adsorbed on MoS$_2$, with one oleamide subchain aligned along the zigzag lattice direction of substrate. The 2D PES exhibits the same hexagonal symmetry as the underlying lattice of MoS$_2$. The energy valleys in *blue* represent the most stable adsorption positions for zigzag aligned oleamide molecules. **b** 1D projection along the $x$ (zigzag) and $y$ (armchair) directions of the same 2D PES in **a**. **c** Assembly of oleamide molecules on MoS$_2$. Each oleamide molecule is adsorbed onto the MoS$_2$ surface at the thermodynamically most stable position, as indicated by the energy valley of the 2D PES in **a**. The *red arrow* on the *left* indicates the orientation of the nanoribbon, which is exactly the zigzag lattice direction of MoS$_2$. The color code of the atoms is Mo: *light blue*, S: *yellow*, C: *gray*, H: *white*, O: *red*, and N: *dark blue*, respectively

molecules and all layers above align along the same direction as the first molecular layer and stack right on top of the molecules in the bottom layers (Supplementary Fig. 13 and Supplementary Note 3). Similar calculations can be performed on other 2D atomic crystals to reveal their respective preferred adsorption directions.

**Universal and nondestructive identification with oleamide.** As both experimental and theoretical studies have confirmed that oleamide molecules can be used to probe the lattice orientations of various 2D atomic crystals, we further demonstrated the applications of this approach to identifying the edge orientations, grain domain and stacking rotations, which all significantly affect the intrinsic properties of these materials. Besides mechanically exfoliated 2D materials, well-aligned oleamide nanoribbons were also observed on chemical vapor deposition (CVD)-grown MoS$_2$ regardless of the number of layers (Fig. 1j). With this approach, we identified that the edges of CVD-grown MoS$_2$ flakes were along zigzag directions as the nanoribbons orientated parallel to the edges of the triangular CVD-grown MoS$_2$ flakes, which is consistent with the STEM imaging (Fig. 1j and Supplementary Fig. 14). In addition to the as-grown edges, the MoS$_2$ edges created by oxygen etching were also identified to be along zigzag directions of MoS$_2$ lattice by using oleamide nanoribbons as probes (Supplementary Fig. 15a and Supplementary Note 4). Moreover, the grain domain in polycrystalline CVD-grown MoS$_2$ and the interlayer rotation angles in multilayer MoS$_2$ can also be easily identified with this approach (Supplementary Fig. 15b, c and Fig. 1j). Therefore, this approach is universal in determining

the lattice orientations of various 2D atomic crystals, regardless of the chemical compositions, preparation methods and geometry.

Besides being universal in identifying different kinds of 2D atomic crystals, the most prominent advantages of our identification approach is that it is simple and nondestructive. This identification process is very simple as only conventional AFM imaging is involved to observe the orientation of oleamide nanoribbons. This process can be further simplified to directly image the orientation of the oleamide assembly with an optical microscope as some of the nanoribbons can grow up to micrometer scale (Fig. 4b, c) as the result of molecule transfer from smaller nanoribbons with higher chemical potential to bigger ones with lower chemical potential, similar to the Ostwald ripening of crystals in a liquid phase (Fig. 4a)[29, 30]. More importantly, this approach is nondestructive as the melting point of oleamide is relatively low (~75 °C) and thus can desorb from the surface of the 2D atomic crystals during storage under ambient conditions (Supplementary Fig. 16). To speed up the desorption, we annealed the MoS$_2$ samples with oleamide assembly in vacuum at 300 °C for 1 h and observed that all the nanoribbons vanished (Figs. 4d, e) without inducing obvious peak position shift in photoluminescence (PL) and Raman spectra (Fig. 4f, Supplementary Fig. 17 and Supplementary Note 5) and the surface roughness of the MoS$_2$ flakes was identical to the pristine value after the removal (0.264 nm (before) and 0.269 nm (after)). To further rule out the possible degradation on the intrinsic properties of MoS$_2$ by the identification process, we fabricated back-gated field effect transistors on both pristine and identified monolayer MoS$_2$ (Supplementary Fig. 18) and compared their electrical performances. The averaged mobility

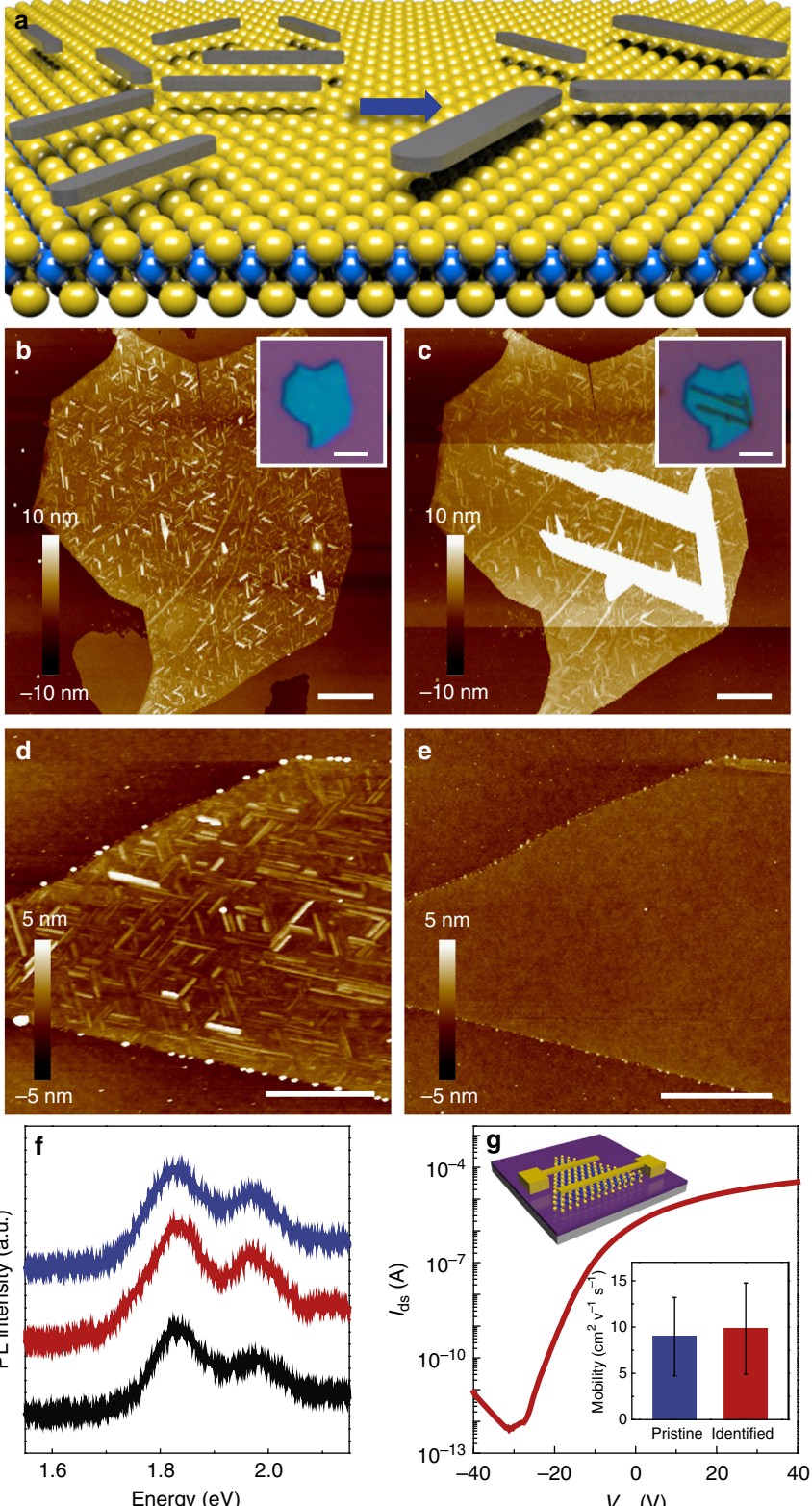

**Fig. 4** Optical visualization and nondestructive removal of oleamide nanoribbons. **a**, Schematic for the evolution of nanoribbons driven by chemical potential. Small nanoribbons shrink while the large ones grow up. **b**, **c** AFM images of oleamide nanoribbons on a few-layer $MoS_2$ flake after 1 h and 8 days under ambient condition, respectively. *Scale bar*, 1 μm. *Inset* Optical images. *Scale bar*, 2 μm. **d**, **e** AFM images of a monolayer $MoS_2$ flake with oleamide nanoribbons before and after annealing in vacuum at 300 °C for 1 h, respectively. *Scale bars*, 500 nm. **f** PL spectra collected from the same monolayer $MoS_2$: pristine (*black*), after oleamide self-assembly (*red*), and after the removal of nanoribbons (*blue*). **g** Typical $I_{ds}$-$V_{gs}$ curve for CVD-grown monolayer $MoS_2$ flakes after the removal of nanoribbons ($V_{ds} = 1$ V). *Inset* The averaged mobility of pristine (*blue*) and identified $MoS_2$ after the removal of nanoribbons (*red*)

of monolayer CVD-grown $MoS_2$ flakes that underwent oleamide assembly and removal is $\sim 8\,cm^2\,V^{-1}\,s^{-1}$, similar to the value obtained from their pristine counterparts (Fig. 4g), indicating that the intrinsic properties of underlying 2D atomic crystals can be well preserved throughout the whole processes.

## Discussion

In summary, we explore the self-assembly of molecules on various 2D atomic crystals by using oleamide molecules as an example both from experimental and theoretical aspects. We show that the periodic atomic arrangements of these materials can guide the assembly of molecules into well-ordered supramolecular structures and correlate the orientations of the oleamide assembly with the lattice structure of the underlying 2D atomic crystals. With self-assembled oleamide nanoribbons as probes, the crystallographic orientations, grain boundaries and stacking configurations of a large variety of 2D atomic crystals regardless of preparation methods and geometry can be easily identified. More importantly, this approach is nondestructive as confirmed by both spectroscopic and electrical measurements, which enables further applications of the identified materials. By using molecules as probes, we open up a new way for simple and nondestructive identification of lattice structures of various substrates especially for 2D atomic crystals, which greatly facilitates both fundamental and application studies of these emerging materials.

## Methods

**Preparation of 2D materials**. The 2D atomic crystals were obtained by mechanical exfoliation of bulk crystals purchased from 2D Semiconductors on $SiO_2/Si$ substrates with Scotch tape, followed by annealing in vacuum at 320 °C for 30 min to remove residual tape (290 °C for $ReS_2$) before the assembly of oleamide. CVD-grown $MoS_2$ flakes were obtained using electrochemical oxidized Mo foils as precursors and the growth was carried out at 650–800 °C[31].

**Self-assembly of oleamide on 2D materials**. The assembly of oleamide was performed by spin coating 7 μL dilute solution (1.65 mmol $L^{-1}$) of oleamide (Sigma-Aldrich, 99%) in chloroform (Alfa Aesar, 99%) on 2D materials at 2400 r.p.m. for 1 min, followed by baking at 60 °C for 30 min, except for black phosphorus. To prevent the oxidation of black phosphorus, the mechanical exfoliation of black phosphorus and the spin coating of oleamide were conducted in a glove box without baking.

**Characterization of nanoribbon/2D materials**. AFM images were captured with Bruker Dimension Icon in ScanAsyst mode. The optical images were taken with an Olympus BX 51 M microscope. Raman and PL spectra were collected with Horiba-Jobin-Yvon Raman system under 532 nm laser excitation with a power of 2 mW. The Si peak at 520.7 $cm^{-1}$ was used for calibration. SHG measurements were conducted under an 820 nm femtosecond laser with excitation power of 0.4 mW (Spectra-Physics® Inspire™ ultrafast OPO system with tunable wavelength of 345–2500 nm, pulse duration of 100 fs and repetition rate of 80 MHz). Linearly polarized excitation light was focused on the sample by an Nikon objective (100X, N.A. = 0.95). SHG signal was detected with a spectrometer (Princeton Instruments, Acton SpectraPro® SP2500 PYLON). Atomically resolved high-angle annular dark-field STEM images were taken with an ARM-200CF (JEOL) transmission electron microscope operated at 200 kV and equipped with double spherical aberration (Cs) correctors. The attainable resolution of the probe defined by the objective pre-field is 78 picometers.

**Data availability**. The data that support the findings of this study are available from the corresponding authors on request.

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

## Acknowledgements

This work was supported by NSFC (Nos. 21273124, 21322303, 51372134, 21573125, 51522201, 11474006, 51522212, 51421002 and 51672307), National Program for Thousand Young Talents of China, Tsinghua University Initiative Scientific Research Program, Strategic Priority Research Program of Chinese Academy of Science (Grant No. XDB 07030200) and Beijing Municipal Science & Technology Commission (No Z161100002116030). We acknowledge Prof. Yanlian Yang and Changliang Liu for discussing the self-assembly details, Prof. Liming Xie and Menghua Cui for the DLS analysis, Qi Zhang for providing CVD-grown $MoS_2$ samples, Yue Li and Yongzan Zheng for storing oleamide and Yan Tang for processing SHG data.

## Author contributions

L.J. and J.W. conceived the ideas. L.J. and D.W. co-supervised this study. J.W., Z.L. and J.Z. performed the experiments. H.Y. and D.W. performed theoretical calculations. L.G. and X.L. performed STEM measurements and analysis. X.Z. and K.L. performed SHG measurements. R.Z. provided detailed instructions on the assembly of oleamide. L.J., J.W., D.W. and H.Y. co-wrote the manuscript. All the authors discussed the results and commented on the manuscript.

## Additional information

**Competing interests:** The authors declare no competing financial interests.

