## [Peer Review File · Nature Communications]

Reviewers' comments:

Reviewer #1 (Remarks to the Author):

This manuscript describes a simple and elegant "non-invasive" method for the detection of the orientation of the symmetry axes of crystalline 2D materials. This method is based on the self-assembly of molecules on the surface and their crystallization. The alignment of the crystals of this particular molecules used, reflects very well the symmetry of the substrate underneath.

I like the manuscript because of its simplicity. The results are convincing. The value of this manuscript is rather the "technical" elegance, than the new scientific insights that are brought. Indeed, it is known that surfaces template the orientation of crystal(lites).

But, to what extent is it really "universal" as claimed? It has been demonstrated only with one molecule...This is my main criticism to the manuscript. Basically, the authors demonstrate a nice trick, but in my opinion, more work is needed to reveal the mechanism of the alignment, and to give insight on the importance of the balance between molecule-molecule and molecule-substrate interactions.

So,

What is the evidence that crystallization happens at the interface, and not in solution? What happens if crystals, formed in solution, are drop cast from solution on the surface?

Modeling was done at the single molecule level as far as the orientational preference is concerned. In general, there are many cases where molecular orientation in the bulk is not the same as on the surface (first molecular layer). Information is needed on the stacking of the molecules in the bulk crystals, and their orientation with respect to the substrate.

I would like the authors to prove that upon thermal treatment all molecules are desorbed from the surface. I believe that upon thermal treatment the 2D material is not affected, but I'm skeptical about the claim that all molecules are desorbed upon thermal treatment.

Reviewer #2 (Remarks to the Author):

Following the questions to referees:

The major claim of the paper is that self-assembled oleamide molecules reveal the crystallographic axes of 2D layered materials (essentially, the title says this). The results presented are novel; though self-assembly and templating by a surface are known in general, this work takes the most topical and interesting 2D layered materials and examines in detail what self-organisation reveals in each example.

As to whether the present results will "influence thinking in the field", the underlying phenomenon is perhaps not a big surprise but the detailed results are extremely clear and will be very useful and so I would say yes, in that this paper presents an idea that many people (including myself!) will immediately go and try with their own samples. An important point here arises at the end of the paper; the self-assembled layers appear to be removable and non-destructive. If the results on a wider range of materials are consistently as good as those shown in Fig 4d,e, then this idea will find very wide use.

The authors use STEM and Raman as reference techniques to identify crystallographic orientation of their 2D layers. In some cases, I would say that Raman is quicker and easier than the self-assembly process (as in the case of BP or ReS₂, where Raman gives the answer in only a few

minutes, whereas the self-assembly requires subsequent AFM) but, in the case particularly of ReS₂, the present results are beautiful - alignment seems completely uni-directional - and this is an exciting result. For example, I have seen incorrect crystallographic analyses from Raman of CVD-grown 2D materials that this technique could have corrected very clearly.

"Is the work convincing?" Yes - the figures speak for themselves and they convince me! Of course I would like to ask the authors some questions: to what extent are these 'typical' regions (rather than 'best') that they show in their images? What areas can be processed with consistent coverage? (so, could this be applied to cm-scale CVD substrates?). Another question, which I guess the authors can't answer yet, is whether this same mechanism works on metallic 2D materials? They have restricted themselves to semiconductors but, of course, there's a lot of interest in slightly different communities in NbSe₂, TaS₂ and so on. These materials are harder to work with - less stable in air, for example, and so obtaining reversible self-assembly might be harder.

Minor concerns: really these are very few but, again related to the question of reversibility, there are no quantitative vertical scales on any of the AFM pictures, so it's hard to judge any vertical dimensions, and this makes it hard to know how clean the surfaces in Fig 4d,e really are after desorption.

If we consider, for example, Fig 5 of the supplementary information, it's clear from Fig 5a that the ReS₂ there is a fairly thick flake; the AFM of the same flake in Fig 5b is therefore covering a large change in height within its false colour scale. Now, is the colour scale the same in Fig 5c? Could the authors somewhere show some quantitative line scans from the AFM moving perpendicular to the nanoribbons so we can see how high the ribbons are? This seems to be a missed opportunity to provide information that will be much appreciated by readers.

Reply to Reviewer 1 and revisions made accordingly:

This manuscript describes a simple and elegant “non-invasive” method for the detection of the orientation of the symmetry axes of crystalline 2D materials. This method is based on the self-assembly of molecules on the surface and their crystallization. The alignment of the crystals of this particular molecules used, reflects very well the symmetry of the substrate underneath.

I like the manuscript because of its simplicity. The results are convincing. The value of this manuscript is rather the “technical” elegance, than the new scientific insights that are brought. Indeed, it known that surfaces template the orientation of crystal.

We are grateful to the reviewer’s positive comments. We have made revisions according to each comment, as summarized below.

- 1. But, to what extend is it really “universal” as claimed? It has been demonstrated only with one molecule. This is my main criticism to the manuscript. Basically, the authors demonstrate a nice trick, but in my opinion, more work is needed to reveal the mechanism of the alignment, and to give insight on the importance of the balance between molecule-molecule and molecule-substrate interactions.*

Reply: Thanks the reviewer for the very insightful comments. We claimed “this approach is universal in determining the lattice orientations of various 2D atomic crystals, regardless of the chemical compositions, preparation methods and geometry” in our manuscript as we have demonstrated that by using oleamide molecules as probe we successfully identified the crystallographic orientations of ~12 different kinds of two-dimensional (2D) materials as we showed in the following figure. We emphasize that our approach is universal because other approaches such as SHG and polarized Raman measurements are limited to specific materials. “Universal” here does not mean that various molecules can be used as probes. We further clarified this point in our revised manuscript.

Figure R1. The lattice orientations of various 2D materials can be identified with our approach. Scale bars: 500 nm.

The reviewer’s comments on the universality of molecules are very insightful. We used oleamide molecules as an example to demonstrate the concept and there are also some other potential candidates. For example, we tested the self-assembly of stearamide on mechanically exfoliated MoS₂ and Alzheimer’s β -amyloid peptide (1-42) on the surface of MoS₂ crystal and observed that they can also form quasi-1D nanoribbons with specific alignments as shown in the following figure. The alignment of the ribbons can be further optimized by carefully tuning the experimental parameters. Therefore, we believe that in addition to oleamide, there are also other choices for probing the lattice

orientation of various 2D materials. We will continue to work on this point to extend the choices of probes.

Figure R2. **a**, The self-assembly of stearamide on mechanical exfoliated MoS₂. **b**, Alzheimer's β-amyloid peptide on MoS₂, also showing anisotropic quasi-1D nanoribbons similar to oleamide. Schematic of stearamide and the sequence of Alzheimer's β-amyloid peptide (1-42) were shown on the top of the AFM images.

For the alignment mechanism of oleamide molecules, we have explored the detailed mechanism by performing theoretical calculations. We agree with the reviewer that the balance between molecule-molecule and molecule-substrate interactions is critical to the formation of well-aligned ribbons as we discussed on page 7 of the manuscript: *“Because the interactions of oleamide molecules with the surface are stronger than the intermolecular interactions between oleamide molecules, the surface lattice will template the formation of nanostructures that cannot be formed in solution. Thus the adsorbate-substrate interactions dictate the assembly of oleamide molecules on the surface of 2D atomic crystals and the tendency to achieve maximum H-bonding between -C=O and -NH₂ groups of adjacent molecules serves as secondary interactions”*. We will continue to enrich our understandings on the mechanism of alignment by investigating the assembly of more molecules on 2D materials.

Revision made: On page 9 of the main text, we changed “Besides being universal” to “Besides being universal in identifying different kinds of 2D atomic crystals”.

2. *What is the evidence that crystallization happens at the interface, and not in solution? What happens if crystals, formed in solution, are drop cast from solution on the surface?*

Reply: Thanks the reviewer for the very insightful comments. To rule out the possible crystallization of oleamide in solution, we measured the particle size of oleamide solution at the concentration (1.65 mmol L⁻¹) we used for the surface-templated assembly with dynamic light scattering (DLS) which is a widely-used approach for determining the size of assembly in solution (*J. Am. Chem. Soc.* 2008, 130, 2780) and did not observe any particles with size of ~10-1000 nm in solutions except for large particles (~10 μm) of impurities from the solvent (Figure R3a, b). For comparison, we also did the same measurements on 5, 10, 15, 20-Tetraphenyl-21H, 23H-porphine cobalt (II) (CoTPP) solution at the same concentration (1.65 mmol L⁻¹) and found that particles with averaged diameter of ~90 nm were well dispersed in the solution (Figure R3c). After that, we spin-coated the oleamide and CoTPP solutions on SiO₂/Si substrates with mechanical exfoliated MoS₂ flakes also at the same conditions and observed dramatically different results as shown in the following Figure R4. Aligned oleamide ribbons were obtained only on MoS₂ flakes and no ribbons were found on the bare substrate. However, randomly distributed CoTPP assemblies formed on both MoS₂ flakes and the substrate. So we can conclude that the assemblies of oleamide form at the interface of MoS₂ and solution and those of

CoTPP form in the solution as CoTPP has stronger intermolecular interactions (*Langmuir* 2012, 28, 6356). These results further confirm that the intermolecular interactions between the oleamide molecules are weak and thus allows for aligning the molecules with suitable substrates.

Figure R3. Size distributions of the particles in 1.65 mmol L⁻¹ solution of oleamide in chloroform (a), pure solvent (b) and 1.65 mmol L⁻¹ solution of CoTPP in chloroform (c). The large particles (~10 μm) observed in the solutions were induced by the impurities from the solvent.

Figure R4. The comparisons between the self-assembly of oleamide and CoTPP on MoS₂. **a**, The self-assembly of oleamide on MoS₂. **b**, The self-assembly of CoTPP on MoS₂.

Revision made: We added the above Figure R3a, b in Supplementary Information as Fig. 6. We added “As we ruled out the possible formation of oleamide assembly in solution through dynamic light scattering (DLS) measurements (Supplementary Fig. 6), we confirmed that the assembly of oleamide was formed on the surface of 2D materials.” on page 4 in the main text. And added “(Supplementary Fig. 6)” on page 7 in the main text after “the surface lattice will template the formation of nanostructures that cannot be formed in solution”.

3. Modeling was done at the single molecule level as far as the orientational preference is concerned. In general, there are many cases where molecular orientation in the bulk is not the same as on the surface (first molecular layer). Information is needed on the stacking of the molecules in the bulk crystals, and their orientation with respect to the substrate.

Reply: Thanks the reviewer for the valuable advice. As discussed above, we ruled out the possible crystallization in solution and confirmed that the ribbons were formed on the surface of atomic crystals. To verify the identical orientation of the bottom and top molecular layers, we modeled the adsorption and assembly of oleamide on the surface of atomic crystals in a layer-by-layer manner. The assembly of 2nd layer oleamide is driven by the inter-layer and intra-layer interactions between oleamide molecules. To investigate the preferred orientation of the 2nd layer of oleamide, we performed the potential energy scan of an oleamide dimer on the 1st molecular layer of nanoribbon with similar computational method that we used to determine the orientation of the 1st layer oleamide on the surface of atomic crystals, as described in Supplementary Information. The rotation (Figure R5a) and translation (Figure R5b, c) potential energy curves show that the oleamide dimer in the 2nd layer prefers to align parallel to the underneath 1st layer oleamide dimer and stacks right on the top of it. Based on the thermodynamically most stable adsorption position of oleamide dimers, we constructed the assembly pattern of oleamide on the 1st molecular layer of nanoribbon, as shown in Figure R5d. In this pattern, the 2nd layer molecules assemble in the same way as the ones in the 1st layer. In other words, the surface lattice of atomic crystals templates the assembly of the 1st layer

oleamide molecules, and the 1st layer of nanoribbon thus formed serves as template for the 2nd layer assembly, and so on. Thus, the 2nd layer molecules and all layers above will align along the same direction as the 1st layer.

Figure R5. Theoretical calculations of the 2nd layer oleamide assembly on the 1st layer of oleamide nanoribbon. **a**, Rotation potential energy curve of an oleamide dimer on top of the single-layered oleamide nanoribbon. The minima at 0° and 180° indicates that the 2nd layer oleamide aligns parallel to the 1st layer ones, and also along the zigzag lattice direction of the substrate. **b**, 2D translation PES of a zigzag aligned oleamide dimer adsorbed on the oleamide nanoribbon. The energy valleys in blue represent the most stable adsorption positions, at which the 2nd layer molecules stack right on the top of the 1st layer ones. **c**, 1D projection along the *x* (zigzag) and *y* (armchair) directions of the same 2D PES in **b**. **d**, Assembly of the 2nd layer oleamide on the 1st layer of oleamide nanoribbon. Each oleamide dimer is adsorbed onto the nanoribbon at the thermodynamically most stable position, as indicated by the energy valley of the 2D PES in **b**. The color code of the atoms is C: cyan (2nd layer) and grey (1st layer), H: white, O: red, and N: dark blue, respectively.

Revision made: In Supplementary Information, we added the above Figure R5 as Fig. 11 and related discussions. We added “The first molecular layer of nanoribbons will in turn template the assembly of the second layer. The rotation and 2D PES scan showed that the oleamide molecules in the second layer and all layers above align along the same direction as the first layer ones and stack right on the top of them (Supplementary Fig. 11).” on page 8 in the main text.

4. *I would like the authors to prove that upon thermal treatment all molecules are desorbed from the surface. I believe that upon thermal treatment the 2D material is not affected, but I'm skeptical about the claim that all molecules are desorbed upon thermal treatment.*

Reply: Thanks the reviewer for pointing out this issue. We admit that it is not feasible to prove the complete desorption of all molecules experimentally. As recognized by the reviewer, the majority of the molecules can be desorbed upon thermal treatment as we displayed in Fig. 4b and 4c of the main text and the intrinsic properties of 2D materials were well reserved. We also analyzed the roughness of pristine MoS₂ and the same flake after the removal of self-assembly and obtained almost identical values of 0.264 nm (before) and 0.269 nm (after), suggesting the efficient removal of oleamide nanoribbons. We modified our claim on the removal of oleamide molecules in the revised manuscript.

Revision made: On page 9, we removed “totally” from “totally desorb from the surface of the 2D atomic crystals during storing under ambient conditions” and we also removed the claim “Moreover, the oleamide nanoribbons can be completely removed from the identified 2D materials...” from the Abstract to make our claim more precise.

We thank the reviewer for his/her very careful review and insightful comments!

Reply to Reviewer 2 and revisions made accordingly:

The major claim of the paper is that self-assembled oleamide molecules reveal the crystallographic axes of 2D layered materials (essentially, the title says this). The results presented are novel; though self-assembly and templating by a surface are known in general, this work takes the most topical and interesting 2D layered materials and examines in detail what self-organisation reveals in each example.

We are grateful to the reviewer's positive comments. We have made revisions according to each comment, as summarized below.

1. *To what extent are these "typical" regions (rather than "best") that they show in their images? What areas can be processed with consistent coverage? (so, could this be applied to cm-scale CVD substrates?)*

Reply: We thank the reviewer for raising this important point. The images we showed in our manuscript are the typical ones, not the best. For the mechanical exfoliated 2D samples, we usually randomly picked several single- or few-layered flakes over $1 \times 1 \text{ cm}^2$ area and carried out AFM imaging after the assembly as we showed in the following figure. The well-aligned ribbons can be found on >90% of the flakes and the flakes without ribbons usually begin with too many Scotch tape residues which prevent the assembly of oleamide on the flakes.

Figure R6. The coverage of oleamide self-assembly on mechanical exfoliated MoS_2 flakes. **a**, The optical image of mechanical exfoliated MoS_2 flakes at a large scale. **b-g**, The AFM images of oleamide self-assembly on MoS_2 flakes randomly selected at different locations on **a**. Scale bars: 1 μm .

For the areas that this approach can be applied to, we would like to say in principle, this approach can be applied to very large area as long as the surfaces are clean enough and the surface atoms are well-packed. We demonstrated the large area processing capability of our approach with the surface of freshly cleaved HOPG ($\sim 1 \times 1 \text{ cm}^2$) which is the largest clean and highly crystalline surface we can obtain. As shown in the following Figure R7, aligned ribbons of oleamide with the same orientations were observed on the HOPG surface over the whole area. The densities and sizes of the ribbons may be varied over a large area as the surface energies and the concentrations of the solution during spin-coating can be varied at different locations but the alignment and orientations of ribbons which are critical for probing the lattice orientations are identical over the whole area. As the preparation of clean and highly crystalline CVD-grown samples at cm-scale is still very challenging and this kind of samples are not available to us, we cannot test our approach on such kind of samples at the current stage. We will definitely try our approach on large area CVD-grown samples as suggested by the reviewer as long as we can get access to suitable samples.

Figure R7. The coverage of oleamide self-assembly on HOPG surface. **a**, The photo of a piece of 1 cm×1 cm HOPG substrate which is divided into 9 regions. **b-j**, AFM images of oleamide self-assembly formed on the HOPG surface in the regions marked with 1-9. Scale bars: 300 nm.

Revision made: To demonstrate that our approach can be applied to a large area we added the above Figure R6 and Figure R7 in Supplementary Information as Fig. 4 and 5. On page 4 of the manuscript, we added “over a large area (Supplementary Fig. 4 and Fig. 5)”.

2. *Another question, which I guess the authors can't answer yet, is whether this same mechanism works on metallic 2D materials? They have restricted themselves to semiconductors but, of course, there's a lot of interest in slightly different communities in NbSe₂, TaS₂ and so on. These materials are harder to work with-less stable in air, for example, and so obtaining reversible self-assembly might be harder.*

Reply: We thank the reviewer for these very insightful comments. To address this point, we tested our approach on several metallic 2D TMDCs, such as NbSe₂, TaS₂ and TiSe₂. As pointed out by the reviewer, these materials are very easy to be oxidized once exposed to air, so we did the exfoliation and assembly in a glove box. As shown in the following figures, well-aligned ribbons are also formed on these samples, confirming that our approach can be also applied to these metallic 2D materials. As concerned by the reviewer, the removal of the assemblies are more difficult on these materials also due to their poor chemical stability. We will continue to work on the assembly and removal of oleamide on these metallic 2D TMDCs and will try to apply this approach to reveal the relationship between the lattice orientations, stacking rotation angles and their phase transition behaviors of these superconducting/CDW materials.

Figure R8. a-c, AFM images of oleamide nanoribbons on mechanically exfoliated metallic TMDCs flakes: **a**, NbSe₂, **b**, TaS₂, **c**, TiSe₂, also showing a three-fold symmetry. Scale bars: 500 nm. **d-e**, Raman spectra collected on the same flakes after self-assembly: **d**, NbSe₂ (A_{1g} mode $\sim 229\text{ cm}^{-1}$ and E_{2g} mode $\sim 236\text{ cm}^{-1}$), **e**, TaS₂ (E_g mode $\sim 100\text{ cm}^{-1}$ and A_{1g} mode $\sim 242, 303$ and 381 cm^{-1}), **f**, TiSe₂ (E_g mode $\sim 135\text{ cm}^{-1}$ and A_{1g} mode $\sim 195\text{ cm}^{-1}$).

Revision made: We added the above Figure R8a-c in Supplementary Information as Fig. 2b-d. On page 4 of the manuscript, we changed “various hexagonal 2D atomic crystals such as mechanical exfoliated graphene, h-BN, MoS₂, MoSe₂, MoTe₂, WS₂, WSe₂ and so on (Fig. 1b-g and Supplementary Fig. 1)” to “various hexagonal 2D atomic crystals such as mechanical exfoliated graphene, h-BN, MoS₂, MoSe₂, MoTe₂, WS₂, WSe₂, NbSe₂, TaS₂ and so on (Fig. 1b-g and Supplementary Fig. 2)”.

3. *Really these are very few but, again related to the question of reversibility, there are no quantitative vertical scales on any of the AFM pictures, so it's hard to judge any vertical dimensions, and this makes it hard to know how clean the surfaces in Fig. 4d,e really are after desorption .*

Reply: Thank the reviewer for pointing out this issue. We have added the vertical scales in Fig. 4b-e. We also analyzed the roughness of pristine MoS₂ and the same flake after the removal of self-assembly and obtained almost identical values of 0.264 nm (before) and 0.269 nm (after), suggesting the efficient removal of oleamide nanoribbons.

Revision made: We added the vertical scales in Fig. 4b-e. On page 10 of the main text, we added “and the surface roughness of the MoS₂ flakes was identical to the pristine value after the removal (before: 0.264 nm and after: 0.269 nm)”.

4. *If we consider, for example, Fig. 5 of the Supplementary information, it's clear from Fig. 5a that the ReS₂ there is a fairly thick flake; the AFM of the same flake in Fig. 5b is therefore covering a large change in height within its false color scale. Now, is the color scale the same in Fig. 5c? Could the authors somewhere show some quantitative line scans from the AFM moving perpendicular to the nanoribbons so we can see how high the ribbons are? This seems to be a missed opportunity to provide information that will be much appreciated by readers.*

Reply: Thank the reviewer for pointing out this issue and the kind suggestions. These two figures are in different color scale. Supplementary Fig. 5c is a zoomed-in AFM image to show oleamide nanoribbons on ReS₂ more clearly. For clarity, we added the scale bar for each figure in the revised Supplementary Information. And as suggested by the reviewer, we added section analysis of several ribbons to show the typical height of the ribbons as shown in the following figure.

Figure R9. Height analysis of several oleamide nanoribbons.

Revision made: We added the above figure in Supplementary Information as Fig. 1. On page 4 of the manuscript, we changed “After the assembly, dense nanoribbons (500 ± 200 nm and 50 ± 20 nm in length and width, respectively) were observed on various hexagonal 2D atomic crystals” to “After the assembly, dense nanoribbons (500 ± 200 nm, 50 ± 20 nm and 5 ± 4 nm in length, width and height, respectively, Supplementary Fig.1) were observed on various hexagonal 2D atomic crystals”.

We thank the reviewer for his/her very careful review and insightful comments!

REVIEWERS' COMMENTS:

Reviewer #1 (Remarks to the Author):

I believe that the authors addressed the comments in an appropriate way. I have only one remark, concerning the way the authors have addressed my question if crystallization may happen in solution rather than at the liquid-solid interface. In the text (rebuttal letter), claims are made that oleylamide does not form crystallites on SiO₂/Si and that the organization of CoTPP crystals on SiO₂/Si is random. I recommend the authors to provide experimental evidence for these claims (so provide AFM images of bare SiO₂/Si, MoS₂ on SiO₂/Si, and CoTPP on SiO₂/Si in supporting information). Moreover, I find the claim that "However, randomly distributed CoTPP assemblies formed on both MoS₂ flakes and the substrate." not very convincing. Looking at Fig. R4b, I'm not convinced that the orientation of CoTPP crystals on MoS₂ is random. Providing a histogram would be useful. These are minor comments though, and don't jeopardize the quality of the manuscript.

Reviewer #2 (Remarks to the Author):

The authors have complied very extensively with the suggestions of both reviewers and have provided significant amounts of new data as well as clarifying unclear points in the original data. Altogether I believe this will be a very useful and interesting paper for a wide readership and publication should now proceed.

Reply to Reviewer 1 and revisions made accordingly:

I believe that the authors addressed the comments in an appropriate way. I have only one remark, concerning the way the authors have addressed my question if crystallization may happen in solution rather than at the liquid-solid interface. In the text (rebuttal letter), claims are made that oleamide does not form crystallites on SiO₂/Si and that the organization of CoTPP crystals on SiO₂/Si is random. I recommend the authors to provide experimental evidence for these claims (so provide AFM images of bare SiO₂/Si, MoS₂ on SiO₂/Si, and CoTPP on SiO₂/Si in supporting information). Moreover, I find the claim that “However, randomly distributed CoTPP assemblies formed on both MoS₂ flakes and the substrate.” not very convincing. Looking at Fig. R4b, I’m not convinced that the orientation of CoTPP crystals on MoS₂ is random. Providing a histogram would be useful. These are minor comments though, and don’t jeopardize the quality of the manuscript.

Reply: Thanks the reviewer for the positive comments and insightful suggestions. We included the AFM images of bare SiO₂/Si, MoS₂ on SiO₂/Si, and CoTPP on SiO₂/Si with MoS₂ in the revised Supplementary Information as suggested by the reviewer (Figure R1). We also followed the advice of the reviewer on providing the histogram of the orientations of CoTPP on MoS₂ (Figure R2). The histograms well supported our claim that the assemblies of CoTPP are randomly distributed on both MoS₂ and the SiO₂/Si substrate.

Figure R1. AFM images of bare SiO₂/Si (a), bare MoS₂ on SiO₂/Si (b), oleamide assemblies on SiO₂/Si with MoS₂ (c) and CoTPP assemblies on SiO₂/Si with MoS₂ (d).

Figure R2. Statistics of the orientations of CoTPP assemblies on MoS₂ and SiO₂/Si. **a,b** AFM image and histogram of the orientations of CoTPP assemblies on MoS₂. **c,d** AFM image and histogram of the orientations of CoTPP assemblies on SiO₂/Si.

Revision made: We added the above Figure R1 and Figure R2 in Supplementary Information as Fig. 6 a-d and Fig. 7. We added “We ruled out the possible formation of oleamide nanoribbons in solution through dynamic light scattering (DLS) measurements (Supplementary Fig. 6) and a control experiment on the self-assembly of 5, 10, 15, 20-tetraphenyl-21H, 23H-porphine cobalt (II) (CoTPP) on MoS₂ (Supplementary Fig. 6 and Supplementary Fig. 7), and therefore, confirmed that oleamide nanoribbons were formed on the surface of 2D materials instead of in solution.” on page 4 in the main text. And added “(Supplementary Fig. 6)” on page 7 in the main text after “the surface lattice will template the formation of nanostructures that cannot be formed in solution”.

Reply to Reviewer 2 and revisions made accordingly:

The authors have complied very extensively with the suggestions of both reviewers and have provided significant amounts of new data as well as clarifying unclear points in the original data. Altogether I believe this will be a very useful and interesting paper for a wide readership and publication should now proceed.

Reply: We are grateful to the reviewer’s positive comments.